# Missing Premise exacerbates Overthinking:
# Are Reasoning Models losing Critical Thinking Skill?

**Chenrui Fan[1]\***, **Ming Li[1]\***, **Lichao Sun[2]**, **Tianyi Zhou[1]**
[1]University of Maryland; [2]Lehigh University
{cfan24, minglii, tianyi}@umd.edu

## Abstract

We find that the response length of reasoning LLMs, whether trained by reinforcement learning or supervised learning, drastically increases for ill-posed questions with missing premises (MiP), ending up with redundant and ineffective thinking. This newly introduced scenario exacerbates the general overthinking issue to a large extent, which we name as the MiP-Overthinking. Such failures are against the "test-time scaling law" but have been widely observed on multiple datasets we curated with MiP, indicating the harm of cheap overthinking and a lack of critical thinking. Surprisingly, LLMs not specifically trained for reasoning exhibit much better performance on the MiP scenario, producing much shorter responses that quickly identify ill-posed queries. This implies a critical flaw of the current training recipe for reasoning LLMs, which does not encourage efficient thinking adequately, leading to the abuse of thinking patterns. To further investigate the reasons behind such failures, we conduct fine-grained analyses of the reasoning length, overthinking patterns, and location of critical thinking on different types of LLMs. Moreover, our extended ablation study reveals that the overthinking is contagious through the distillation of reasoning models' responses. These results improve the understanding of overthinking and shed novel insights into mitigating the problem. Our code and data can be found in: https://github.com/tianyi-lab/MiP-Overthinking.

## 1 Introduction

Reasoning abilities in large language models (LLMs) have become a cornerstone of advanced AI applications (Huang & Chang, 2023; Li et al., 2025b; Ahn et al., 2024; Wang et al., 2025), powering breakthroughs in mathematical reasoning (Xiong et al., 2025; Xia et al., 2025), code generation (Liu et al., 2024), and commonsense question answering (Wang & Zhao, 2023). These gains often stem from the scaling law of model/dataset sizes (Kaplan et al., 2020) in both pre-training (Shao et al., 2024) and post-training, which unlock emergent capabilities such as step-by-step reasoning and reflection skills witnessed on OpenAI's GPT-o1 (OpenAI, 2024b) and the open-source DeepSeek-R1. By leveraging supervised fine-tuning (SFT) on expert responses (Li et al., 2025b; Ye et al., 2025; Muennighoff et al., 2025) and/or reinforcement learning (RL) (DeepSeek-AI et al., 2025), these models are tailored to produce detailed multi-step reasoning paths, whose length increase usually associated with improved performance on complex tasks such as math reasoning and programming.

Despite the fascinating reasoning capabilities exhibited on recent models, there is growing concern about the efficiency and quality of the long reasoning process (Sui et al., 2025). Chen et al. (2025b) first raises the "overthinking" problem in reasoning LLMs, which is reflected by the excessively long reasoning paths generated for extremely simple queries. For example, even for questions like "*What is the answer of 2 plus 3?*", existing reasoning models might generate hundreds of response tokens.

In particular, the ill-posed queries are unsolvable due to the lack of a necessary premise or condition. We call the reasoning failure for the ill-posed queries **Overthinking under**

---

*Equal Contribution.

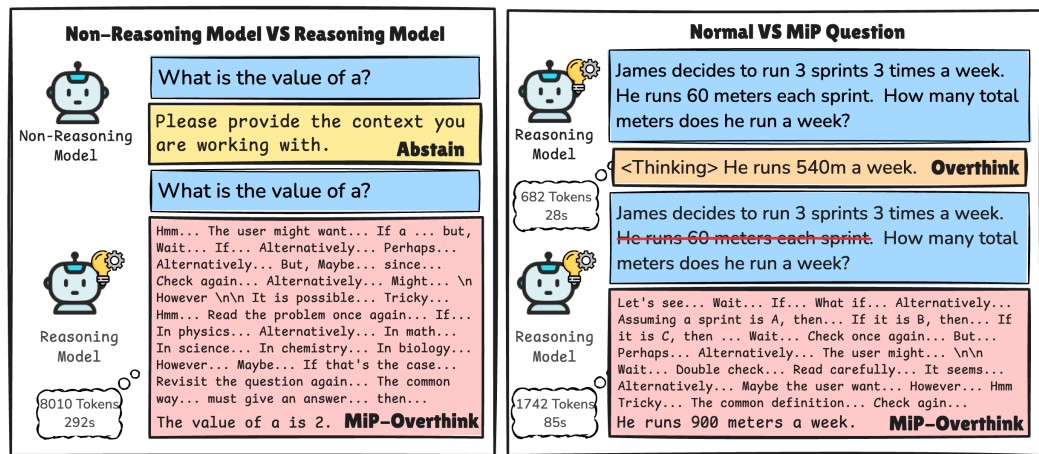

Figure 1: Illustration of MiP-Overthinking. When queried by questions with missing premises, the response length of reasoning models increases excessively, and they cannot abstain from answering with MiP identified. The left shows a query with an undefined variable, while the right compares a well-defined GSM8K question with its MiP variant. Reasoning models' responses to MiP questions are much longer than those for well-defined questions and those generated by non-reasoning models. The left corner of each response report the response length and thinking time by DeepSeek-R1.

**Missing Premise (MiP-Overthinking).** For example, the simplest MiP question is *What is the value of a?* , as shown on the left part of Figure 1. Without providing any other information regarding $a$, it is evidently unsolvable. However, DeepSeek-R1 generates thousands of tokens and spends several minutes thinking about this question before outputting the final meaningless answer [1]. In this paper, we find that a trivial type of ill-posed queries will significantly exacerbate the overthinking of reasoning models, resulting in excessively redundant and meaningless thinking. In contrast, humans and even non-reasoning models are often immune to such scenarios and quickly end up by questioning the validity of the given query, indicating the critical thinking capability. This exposes a risk of the abuse of thinking patterns and a lack of critical thinking on the models trained for deep thinking. Ideally, a model with critical thinking skills is expected to identify the missing premise and quickly respond by either requesting clarification or gracefully indicating that it cannot proceed (Cole et al., 2023; Amayuelas et al., 2024).

MiP-Overthinking differs from the widely discussed overthinking issue (Cuadron et al., 2025), in which the query is usually well-defined, but a model applies much more reasoning than necessary for little benefit. MiP-Overthinking, by contrast, happens when the question itself is ill-posed and lacks sufficient information to be solved. For example, the right of Figure 1 presents a well-defined question from GSM8K and a MiP variant, where the latter triggers a drastic increase of the generated tokens on recent reasoning models compared with the general overthinking. **Overthinking can be presented by the length difference between models addressing the same well-defined questions, while MiP-Overthinking can be presented by the additional tokens generated due to MiP.** It further reveals the lack of critical thinking that questions the validity of ill-posed questions and quickly identifies MiP, thus abstaining from answering the questions. Moreover, we observe that reasoning models' ineffective and redundant thinking often cannot stop even after successful notice of MiP, violating the expectation of test-time scaling law. Hence, MiP-Overthinking indicates potential drawbacks of current training recipes of reasoning models.

To systematically investigate this issue, we construct a suite of MiP questions designed to trigger the overthinking failures in a controlled way. These include synthetic questions generated by Rule-based Formula (queries where a formula reference is empty or nonsensical) and careful modifications of established datasets across diverse levels of difficulties, including SVAMP, GSM8K, and MATH500. On the modified datasets of MiP questions,

---

[1]The screenshot on how DeepSeek-R1 responds to this question is shown in the Appendix D.

we empirically evaluate a wide range of state-of-the-art LLMs, from reasoning models to non-reasoning models and from open-sourced models to proprietary models, to ensure the generalizability of our findings. Our analysis is mainly based on three evaluation metrics, the length of generated responses, the accuracy on well-defined questions, and the abstain rate on ill-posed questions with MiP.

**Main Contributions:** We present the first in-depth study of *Overthinking under Missing Premise (MiP-Overthinking)*, which reveals a critical shortcoming in existing reasoning models: Although they appear to follow coherent reasoning patterns, **they lack genuine critical thinking capabilities**. To systematically analyze this issue, we curate four MiP datasets covering various difficulty levels and three ill-posed question generation strategies, i.e., *Rule-Based Generation, Body-Question Swapping, and Essential-Premise Removal.* We then evaluate a wide range of large language models including reasoning-based and non-reasoning ones. Our empirical results illuminate the differences in how models handle well-defined vs. MiP questions, ultimately offering insights into the limitations of existing reasoning models.

**Our key findings:**

1. Missing premise in questions induces reasoning models to generate significantly longer ($2\times$ to $4\times$ more tokens) responses than general overthinking on well-defined questions. The increased tokens fail to help identify MiP in the ill-posed questions, surprisingly **contradicting the widely-discussed test-time scaling law**.

2. In contrast, given MiP questions, **non-reasoning models generate consistently shorter responses and quickly identify MiP**, demonstrating greater robustness to the absence of critical information.

3. Reasoning models respond differently to well-defined vs. MiP questions: they mostly follow stable chain-of-thoughts for the former, but are often **trapped in a self-doubt loop, repeatedly revisiting the question, and guessing the user intentions** under MiP, resulting in an explosion of tokens.

4. Reasoning models often can **notice the existence of MiP or identify it at an early stage**, but they **hesitate to commit to this judgment** and keep outputting ineffective thinking.

## 2 Missing Premise Definition and Construction

### 2.1 Definition of Missing Premise

Prior to introducing the construction our dataset and analyzing the behavior of reasoning models on problems with missing premises, we formally define the Missing Premise (MiP) problem to establish a rigorous foundation for our subsequent analysis.

---

**Definition 1** (Missing Premise Problem). *Let $\mathcal{Q}$ be a question, and let $P = \{P_1, \ldots, P_n\}$ be a set of premises. Define the function mapping premises and a question to the set of logically valid answers as:*

$$\mathcal{F}(P, \mathcal{Q}) = \{A \mid P \vdash A, A \text{ is an answer resolving } \mathcal{Q}\} \tag{1}$$

*where $\vdash$ denotes logical entailment. Consider a proper subset $P' = P \setminus \{P_i\}$ for some $P_i \in P$. The tuple $(P', \mathcal{Q})$ forms a **missing premise problem** if and only if:*

$$|\mathcal{F}(P, \mathcal{Q})| = 1 \quad and \quad |\mathcal{F}(P', \mathcal{Q})| \neq 1 \tag{2}$$

*This indicates that the removed premise $P_i$ is essential for uniquely determining the logically valid answer to the question $\mathcal{Q}$.*

---

According to Definition 1, an ideal reasoning system should efficiently identify the absence of a critical premise and terminate its inference process upon recognizing that the available information is insufficient to derive a unique solution to the given problem. However, our empirical analysis in Section 3.2 demonstrates that state-of-the-art reasoning models

| Dataset | Example | Diff | Count | Pair | Method |
|---------|---------|------|-------|------|--------|
| MiP-Formula | What is the value of $\ln(a + b)$? | $\star$ | 50 | ✗ | Rule-Based Generation |
| MiP-SVAMP | Paco had 26 salty cookies and 17 sweet cookies. He ate 14 sweet cookies and 9 salty cookies. ~~How many salty cookies did Paco have left?~~ How many pencils does she have? | $\star$ | 300 | ✗ | Body-Question Swapping |
| MiP-GSM8K | James decides to run 3 sprints 3 times a week. ~~He runs 60 meters each sprint.~~ How many total meters does he run a week? | $\star\star$ | 582 | ✓ | Essential-Premise Removal |
| MiP-MATH | There are 360 people in my school. 15 take calculus, physics, and chemistry, and 15 don't take any of them. 180 take calculus. Twice as many students take chemistry as take physics. 75 take both calculus and chemistry, and 75 take both physics and chemistry. ~~Only 30 take both physics and calculus.~~ How many students take physics? | $\star\star\star$ | 58 | ✓ | Essential-Premise Removal |

Table 1: Statistics and examples of our curated MiP datasets. For GSM8K and MATH, a premise is removed from the original questions (crossed out) to create MiP questions. *Diff* represents the (estimated) difficulty for models to identify MiP. *Count* denotes the number of questions in the subset. *Pair* indicates whether each MiP question is associated with a well-defined question. *Method* indicates the method used to generate the MiP question.

consistently fail to exhibit this capability. Instead, these models engage in extensive, redundant reasoning chains that consume significant computational resources without ultimately identifying the missing premise.

### 2.2 Overview of Data Construction

To systematically investigate this MiP-Overthinking issue, we construct a suite of MiP questions in a controllable manner. Our MiP questions are sourced from 3 math datasets across different difficulties. In addition, we also construct a synthetic dataset consisting of formulas with unassigned variables. Our ill-posed question generation employs three distinct methods covering three difficulty levels and three strategies to create MiP questions: **Rule-Based Generation**, **Body-Question Swapping**, and **Essential-Premise Removal**.

Then, we further construct 4 MiP datasets utilizing the above methods: **MiP-Formula**, **MiP-SVAMP**, **MiP-GSM8K**, and **MiP-MATH**. For comprehensive implementation details and additional methodological considerations, we refer readers to Appendix B.

## 3 Overthinking under Missing Premise

### 3.1 Evaluation Metrics

To systematically evaluate model responses under MiP, we conduct experiments with a diverse set of reasoning and non-reasoning models. For each model, we analyze calculate the following metrics for the responses across different datasets: **Response Length**, **Abstain Rate for MiP Question**, and **Accuracy for Well-defined Question**. For datasets without reference answers (MiP-Formula and MiP-SVAMP), we only calculate the abstain rate for the questions. Response evaluation is performed using GPT-4o as an automated evaluator. Detailed experimental procedures and evaluation protocols are provided in Appendix A.

### 3.2 Main Results

Figure 2 compares average response length, accuracy on well-defined questions, and the abstain rate on MiP questions across a range of state-of-the-art LLMs, revealing several significant patterns in model behavior.

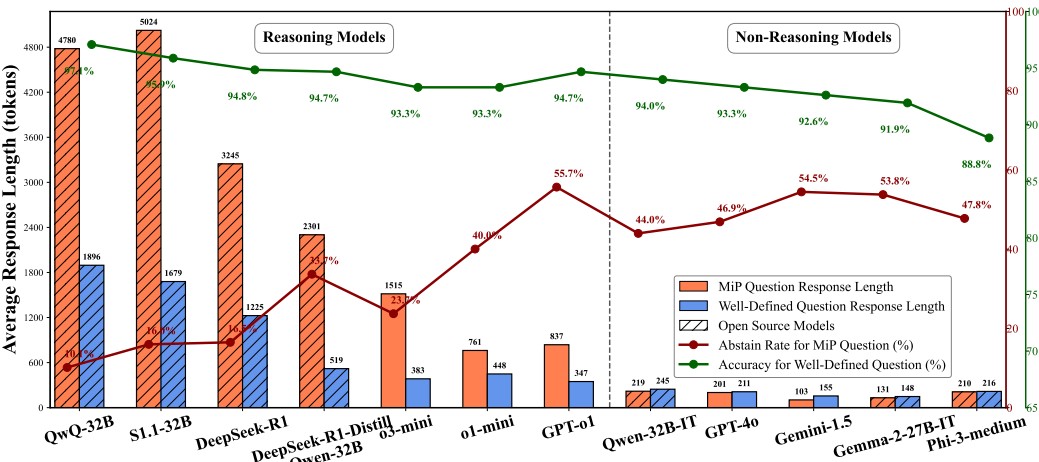

Figure 2: Response lengths, accuracy on well-defined questions, and abstain rate of reasoning/non-reasoning models on MiP questions from our MiP-GSM8K dataset. (1) Existing reasoning models generate significantly longer responses for MiP questions than well-defined questions, while non-reasoning models generate responses of similar lengths for both types of questions, indicating **MiP-Overthinking** for reasoning models. (2) For both questions, reasoning models generate longer responses than non-reasoning models, indicating **General Overthinking**. (3) Although the longer responses by reasoning models slightly improve the accuracy for well-defined questions, it does not enhance the abstain rate for MiP questions, indicating a **contradiction on the test-time scaling law**.

Firstly, existing reasoning models (left side of the figure) display an explosive increase in response length when facing the MiP questions, often producing $2-4\times$ more tokens than general overthinking on well-defined questions. For example, QwQ-32B (Team, 2025) and DeepSeek-R1 (DeepSeek-AI et al., 2025) exhibit a substantial increase from already long reasoning paths on well-defined questions (approximately $1,000$ tokens for simple GSM8K questions) to highly lengthy outputs (more than $3,000$ tokens) under missing premise conditions. On the contrary, no similar issues exist for non-reasoning models (right side of the figure), which generate similar token counts for both types of well-defined and MiP questions. This phenomenon directly illustrates the **MiP-Overthinking** phenomenon as introduced in the paper.

Secondly, comparing the token lengths on well-defined questions between the reasoning and non-reasoning models, reasoning models tend to produce longer responses, even for simple questions, than non-reasoning models, underscoring the inefficient and verbose responses of existing reasoning models. For example, for the non-reasoning models, it only takes approximately 200 tokens for them to generate the responses for well-defined questions, while taking $1,000$ tokens for DeepSeek-R1 and $1,800$ tokens for QWQ-32B to answer the exactly same questions. However, the explosive increase in extra tokens does not lead to corresponding large accuracy improvements, shown in the green line, highlighting the issue of the **General Overthinking**.

Finally, the abstain rates (red line) on MiP questions reveal that although some reasoning models (e.g., GPT-o1) have promising capabilities in abstaining from the MiP questions, most of the other reasoning models are not able to abstain from the given MiP questions correctly despite the dramatically long reasoning paths. This phenomenon indicates that although most existing reasoning models have thinking and reasoning capabilities to some extent, they **lack the critical thinking capabilities** to "reject" ill-posed questions. By contrast, non-reasoning models, though they are not explicitly trained for reasoning, tend to strike a better balance, generating shorter answers that are more likely to acknowledge MiP when the question is ill-posed. This phenomenon reveals a surprising **contradiction on test-time scaling law**.

| Model | Type | MiP-Formula Length↓ | Abstain↑ | MiP-SWAMP Length↓ | Abstain↑ | Type | MiP-GSM8K Length↓ | Abstain↑ | MiP-MATH Length↓ | Abstain↑ |
|---|---|---|---|---|---|---|---|---|---|---|
| **Non-Reasoning Models** | | | | | | | | | | |
| Qwen2.5-32B-Instruct | MiP | 285 | 44.0 | 128 | 98.3 | MiP | 219 | 44.0 | 525 | 15.4 |
| | | | | | | Well-defined | 246 | 0.5 | 1114 | 1.9 |
| GPT-4o | MiP | 338 | 70.0 | 122 | 96.3 | MiP | 202 | 46.9 | 487 | 15.4 |
| | | | | | | Well-defined | 212 | 0.5 | 472 | 1.9 |
| Gemini 1.5 | MiP | 453 | 20.0 | 52 | 99.0 | MiP | 103 | 54.5 | 568 | 5.8 |
| | | | | | | Well-defined | 156 | 0.5 | 502 | 0.0 |
| Gemma-2-27B-IT | MiP | 204 | 85.7 | 89 | 92.0 | MiP | 131 | 53.8 | 338 | 38.5 |
| | | | | | | Well-defined | 148 | 0.3 | 305 | 11.5 |
| Phi-3-medium-128k | MiP | 1465 | 48.0 | 125 | 98.7 | MiP | 210 | 47.8 | 427 | 23.1 |
| | | | | | | Well-defined | 216 | 1.0 | 1549 | 3.8 |
| Qwen3-32B (Non-Reason) | MiP | 496 | 90.0 | 184 | 100.0 | MiP | 279 | 32.6 | 1571 | 17.3 |
| | | | | | | Well-defined | 237 | 1.0 | 1287 | 0.0 |
| **Reasoning Models** | | | | | | | | | | |
| GPT-o1 | MiP | 1123 | 78.0 | 581 | 99.0 | MiP | 838 | 55.7 | 4189 | 30.8 |
| | | | | | | Well-defined | 348 | 0.3 | 2502 | 0.0 |
| GPT-o1mini | MiP | 958 | 66.0 | 639 | 96.7 | MiP | 762 | 40.0 | 2193 | 25.0 |
| | | | | | | Well-defined | 449 | 1.2 | 1913 | 0.0 |
| GPT-o3mini | MiP | 1025 | 76.0 | 1299 | 93.0 | MiP | 1516 | 23.7 | 3772 | 11.5 |
| | | | | | | Well-defined | 384 | 1.4 | 1553 | 0.0 |
| DS Distill Qwen2.5-32B | MiP | 12911 | 42.0 | 921 | 88.3 | MiP | 2302 | 24.6 | 9876 | 5.8 |
| | | | | | | Well-defined | 519 | 0.2 | 3246 | 0.0 |
| DeepSeek R1 | MiP | 4757 | 6.0 | 1996 | 84.3 | MiP | 3246 | 16.5 | 7268 | 3.8 |
| | | | | | | Well-defined | 1226 | 0.2 | 3200 | 1.9 |
| S1.1-32B | MiP | 5284 | 18.0 | 3358 | 57.0 | MiP | 5024 | 16.0 | 9322 | 15.4 |
| | | | | | | Well-defined | 1896 | 0.2 | 5037 | 0.0 |
| QwQ-32B | MiP | 7937 | 0.0 | 3487 | 56.3 | MiP | 4780 | 10.1 | 10242 | 1.9 |
| | | | | | | Well-defined | 1896 | 0.2 | 5037 | 0.0 |
| Qwen3-32B (Reason) | MiP | 5293 | 34.0 | 1872 | 58.0 | MiP | 3149 | 22.4 | 9468 | 5.7 |
| | | | | | | Well-defined | 1723 | 0.0 | 5555 | 0.0 |

Table 2: Comparing response length and abstain rate across different MiP datasets. Shorter lengths and higher abstain rates are preferred. For each column, the top-3 preferred values are colored in green, otherwise red. **MiP-Overthinking, reflected by longer response with low abstain rate, is commonly observed on most existing reasoning models across all datasets, indicating a critical drawback of existing reasoning models.**

Moreover, Table 2 further presents the comparisons on length and abstain rate on other MiP datasets we curated. The preferred results are colored green (shorter responses and higher abstain rate for MiP questions), and the worse results are colored red, from which we can easily discover that reasoning models are prone to generate long responses while having low abstain rates across all datasets, indicating the consistent MiP Overthinking issue of existing reasoning models. In addition, by comparing the behaviors of models on different datasets, we can observe that for the relatively harder dataset (MiP-MATH), all models generate relatively longer responses and obtain lower abstain rates, indicating that harder MiP questions require reasoning capabilities.

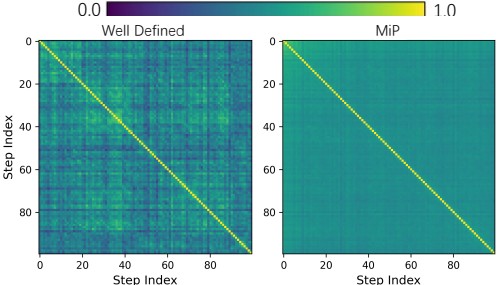

Figure 3: The step-level similarity heatmaps for s1.1 responses towards well-defined (left) and MiP (right) questions in MiP-GSM8K dataset. To avoid differences in matrix size, we only consider responses with more than 50 steps and visualize the average simialrity matrix across first 50 steps. **The heatmap for MiP questions has a higher averaged similarity and lower standard variance, also shown in the heatmap, which indicates the considerable redundancy in its content when responding to MiP questions.**

### 3.3 Thinking Patterns through Tokens

To gain deeper insight into the MiP-Overthinking issue, we compare the reasoning-related token distribution (Lin et al., 2023; Li et al., 2025c) on the MiP-GSM8K dataset. As shown in Table 3, we break down the average usages of several token patterns related to the thinking process, as well as the number of steps for each model to solve the given questions. Specifically, values of *alternatively*, *wait*, *check*, and *but* can be directly

| Models | Type | Alternatively Cnt. | Δ | Wait Cnt. | Δ | Check Cnt. | Δ | But Cnt. | Δ | Hypothesis Cnt. | Δ | Step Cnt. | Δ |
|---|---|---|---|---|---|---|---|---|---|---|---|---|---|
| **Non-Reasoning Models** | | | | | | | | | | | | | |
| Qwen2.5-32B | MiP | 0.0 | 0.0 | 0.0 | 0.0 | 0.0 | 0.0 | 0.3 | 0.2 | 0.0 | 0.0 | 4.3 | -1.3 |
| | Well-defined | 0.0 | | 0.0 | | 0.0 | | 0.1 | | 0.0 | | 5.6 | |
| GPT-4o | MiP | 0.0 | 0.0 | 0.0 | 0.0 | 0.0 | 0.0 | 0.3 | 0.2 | 0.0 | 0.0 | 4.7 | -1.5 |
| | Well-defined | 0.0 | | 0.0 | | 0.0 | | 0.1 | | 0.0 | | 6.2 | |
| Gemini 1.5 | MiP | 0.0 | 0.0 | 0.0 | 0.0 | 0.0 | 0.0 | 0.1 | 0.1 | 0.0 | 0.0 | 1.6 | -2.2 |
| | Well-defined | 0.0 | | 0.0 | | 0.0 | | 0.0 | | 0.0 | | 3.8 | |
| Gemma-2-27B | MiP | 0.0 | 0.0 | 0.0 | 0.0 | 0.0 | 0.0 | 0.1 | 0.1 | 0.0 | 0.0 | 5.2 | -0.5 |
| | Well-defined | 0.0 | | 0.0 | | 0.0 | | 0.0 | | 0.0 | | 5.7 | |
| **Reasoning Models** | | | | | | | | | | | | | |
| DS-Distill Qwen | MiP | 11.5 | 11.4 | 19.7 | 19.3 | 1.0 | 0.8 | 40.1 | 39.3 | 38.4 | 38.0 | 54.9 | 42.2 |
| | Well-defined | 0.1 | | 0.4 | | 0.2 | | 0.8 | | 0.4 | | 12.7 | |
| DeepSeek R1 | MiP | 16.9 | 15.2 | 14.4 | 10.9 | 3.8 | 1.3 | 49.4 | 42.1 | 44.7 | 40.4 | 54.2 | 33.0 |
| | Well-defined | 1.7 | | 3.5 | | 2.5 | | 7.3 | | 4.3 | | 21.2 | |
| S1.1 | MiP | 42.0 | 38.0 | 21.9 | 15.9 | 5.5 | 2.5 | 87.2 | 74.1 | 84.8 | 77.0 | 79.9 | 50.9 |
| | Well-defined | 4.0 | | 6.0 | | 3.0 | | 13.1 | | 7.8 | | 29.0 | |
| QwQ | MiP | 47.0 | 40.3 | 19.4 | 13.0 | 5.0 | 1.6 | 66.1 | 54.2 | 94.1 | 81.7 | 97.9 | 58.7 |
| | Well-defined | 6.7 | | 6.4 | | 3.4 | | 11.9 | | 12.4 | | 39.2 | |

Table 3: Comparisons of reasoning-related token counts on MiP-GSM8K dataset. *Hypothesis* category includes several key words, including *perhaps*, *maybe*, and *might*. *Step* represents the step counts, spited by $\backslash n\backslash n$, where negative values are colored in green and positive in red. Δ denotes the difference between MiP and well-defined questions. **When facing MiP questions, reasoning models encounter explosive growths on reasoning-related tokens and steps, indicating a severe abuse of thinking patterns, while non-reasoning models use fewer steps for MiP questions than well-defined ones.**

counted from the model responses, including the thinking paths of reasoning models. *Hypothesis* category includes several key words, including *perhaps*, *maybe*, and *might*. *Step* represents the step counts, spited by $\backslash n\backslash n$.

Reasoning models exhibit much higher occurrence of tokens such as *alternatively*, *wait*, and *check*, compared with non-reasoning models, whose frequencies remain close to zero, indicating their advanced thinking capabilities. However, when moving from well-defined to MiP questions, reasoning models encounter explosive growths on reasoning-related tokens, indicating a large redundancy in thinking patterns. Moreover, when comparing the changes of steps, reasoning models exhibit a large increase in step count for MiP questions, while non-reasoning models typically show fewer steps, suggesting they quickly conclude the question is unanswerable. With this gap, together with the consistently better abstain rates of the non-reasoning models, we conclude that **the lengthy reasoning steps are mostly redundant and indicate self-doubt thinking patterns for reasoning models.**

## 3.4 Step-level Similarities

To further assess how redundant the generated content becomes under MiP conditions, we examine the step-level similarity within the model's responses on our MiP-GSM8K dataset. Specifically, we divide each response into discrete steps, split by $\backslash n\backslash n$, and compute pairwise cosine similarity scores with embeddings generated by "all-MiniLM-L6-v2" (Reimers & Gurevych, 2019). The visualization is shown in Figure 3, where each value in the heatmap metrix represents the averaged cosine similarities between the corresponding step index. The average similarity score for well-defined question is **0.45** and **0.50** for MiP response. The variance is **7.9e-3** and **8.2e-4** respectively.

As shown in the figure, responses to MiP questions have greater overall similarity across steps and lower standard variance, indicating the considerable redundancy in the content. This means, in many instances, **the model revisits similar partial reasoning or repeats previous sentences with only minor changes, showing a potential self-trapping issue.** Together, these patterns confirm that MiP questions induce a high degree of repetitive

| Model | Type | MiP-Formula | | MiP-SWAMP | | Type | MiP-GSM8K | | MiP-MATH | |
|---|---|---|---|---|---|---|---|---|---|---|
| | | Length↓ | Abstain↑ | Length↓ | Abstain↑ | | Length↓ | Abstain↑ | Length↓ | Abstain↑ |
| **Non-Reasoning Models** | | | | | | | | | | |
| **Qwen3-1.7B (Non-Reason)** | MiP | 437 | 64.0 | 265 | 77.0 | MiP | 331 | 22.2 | 952 | 11.5 |
| | | | | | | Well-defined | 264 | 0.6 | 1235 | 0.0 |
| **Qwen3-8B (Non-Reason)** | MiP | 410 | 94.0 | 246 | 98.0 | MiP | 337 | 40.5 | 1941 | 17.3 |
| | | | | | | Well-defined | 256 | 0.5 | 1538 | 0.0 |
| **Qwen3-32B (Non-Reason)** | MiP | 496 | 90.0 | 184 | 100.0 | MiP | 279 | 32.6 | 1571 | 17.3 |
| | | | | | | Well-defined | 237 | 1.0 | 1287 | 0.0 |
| **Reasoning Models** | | | | | | | | | | |
| **Qwen3-1.7B (Reason)** | MiP | 4986 | 34.0 | 3072 | 30.0 | MiP | 4000 | 10.2 | 9272 | 5.9 |
| | | | | | | Well-defined | 2538 | 0.0 | 5776 | 0.0 |
| **Qwen3-8B (Reason)** | MiP | 5483 | 58.0 | 2656 | 49.4 | MiP | 3851 | 31.3 | 9933 | 4.0 |
| | | | | | | Well-defined | 2620 | 2.0 | 5895 | 1.9 |
| **Qwen3-32B (Reason)** | MiP | 5293 | 34.0 | 1872 | 58.0 | MiP | 3149 | 22.4 | 9468 | 5.7 |
| | | | | | | Well-defined | 1723 | 0.0 | 5555 | 0.0 |

Table 4: Comparing the effects of model sizes and reasoning capabilities within the Qwen3 family. MiP-Overthinking is widely observed among models of different sizes consistently, and there is no consistent pattern between size and severity. **This phenomenon indicates that MiP-Overthinking can not be mitigated by simply scaling up the model size.**

| Metrics | MiP-Formula (Mix) | | | | MiP-SVAMP (Mix) | | | | MiP-GSM8K (Mix) | | | |
|---|---|---|---|---|---|---|---|---|---|---|---|---|
| | Recall (%) | Precision (%) | Time (seconds) | Length (words) | Recall (%) | Precision (%) | Time (seconds) | Length (words) | Recall (%) | Precision (%) | Time (seconds) | Length (words) |
| | 100 | 100 | 5.21 | 20.3 | 98 | 100 | 17.66 | 34.6 | 94 | 96 | 19.80 | 42.5 |

Table 5: Human evaluation on mixed datasets that contain MiP and solvable questions. Humans reliably detect MiP questions while spending only a small amount of time per question. **This observation highlights the significant gap between the current models and human critical-thinking behavior when facing ill-posed tasks.**

content in reasoning models. Rather than terminating early to conclude for insufficient premise, the models fill their reasoning paths with repetitive re-checks and reiterations, significantly inflating token usage without improving real abstain rates.

### 3.5 Effects of Model Sizes and Reasoning Capabilities

To investigate how the model sizes and reasoning capabilities affect the MiP-Overthinking issue within the same model family, we conduct controlled experiments on Qwen3 family models as shown in Table 4. The *Reasoning* and *Non-Reasoning* represent whether to utilize the reasoning mode for the Qwen3 models. As shown in the table, the overall obersvations are consistent with the results in Figure 2, indicating the generalization of this issue, Moreover, by comparing this overthinking phenomenon across different model sizes within the same model family, we observe that the MiP-Overthinking issue occurs consistently and there is no consistent patterns that are strongly correlated with the model size. This observation especially highlights the significance of the MiP-Overthinking issue, which can not be mitigated by simply scaling up the model size.

## 4 Further Discussion

### 4.1 How Humans React to MiP Questions?

To evaluate how humans respond to MiP questions, we conducted a small-scale user study with three graduate-level participants. The participants were presented with mixed versions of the MiP-Formula, MiP-SVAMP, and MiP-GSM8K splits, each consisting of 50 MiP problems and 50 ordinary solvable questions[2]. For every question they were told to decide whether each question is solvable or not and the time spent and the question length in words are recorded for further analysis.

---

[2]The setting is slightly different from LLM's MiP-Overthinking setting, which provides only the non-solvable questions. In the setting, even if the participants are not told that some questions might be unsolvable, they can figure this out and quickly annotate all the questions as non-solvable accordingly. Thus, we utilize the mixed version to avoid this issue.

| Model | MiP-Formula | | | | MiP-GSMR | | | |
|---|---|---|---|---|---|---|---|---|
| | DeepSeek-R1 | DS-Qwen | QwQ | S1.1 | DeepSeek-R1 | DS-Qwen | QwQ | S1.1 |
| In-Process Suspicion Rate | 100% | 100% | 100% | 100% | 95.5% | 83.3% | 99.6% | 100% |
| In-Process First Suspicion Index | 1.32 | 1.36 | 1.42 | 1.16 | 2.01 | 3.90 | 1.77 | 1.61 |

Table 6: The in-process insufficiency suspicion information across different reasoning models on MiP-Formula and MiP-GSMR datasets. The in-process insufficiency suspicion is defined as when the reasoning model suspects the given question is unsolvable during its thinking process. *In-Process Suspicion Rate* represents how many percent of the samples trigger the in-process suspicion. *First Suspicion Index* is the averaged step index where the model first suspects the question's validity. **Most reasoning models do notice the existence of MiP at the very early steps, but they still suffer from low abstain rate and cannot confidently stop the thinking.**

As shown in Table 5, humans achieve near-perfect MiP identification: recall of $\geq 94\%$ and precision of $\geq 96\%$ across all three datasets, while requiring only a few seconds for short algebra problems (MiP-Formula) and under twenty seconds for longer word problems (MiP-SVAMP/GSM8K). This performance is substantially better than the abstain rates of state-of-the-art reasoning LLMs reported in Table 2, highlighting a pronounced gap between current models and human critical-thinking behavior when facing ill-posed tasks. **This observation further highlights the significant gap between the current models and human critical-thinking behavior when facing ill-posed tasks.**

### 4.2 Do Models Know Premises are Missing?

To investigate whether reasoning models recognize the potential unsolvability of questions during their reasoning process, we conducted a detailed analysis of their reasoning chains. We segmented each reasoning chain into discrete steps using $\backslash n \backslash n$ as delimiters and performed step-wise verification to detect whether models express doubt on the question solvability. We introduce two key metrics for this analysis: **In-Process Suspicion Rate**, which measures the percentage of responses where the model expresses doubt about solvability during reasoning, and **First Suspicion Index**, which captures the average step number at which the model first suspects the missing premise. To ensure robust evaluation, we employed GPT-4o to assess each step three times, using majority voting for our final step-level result. The quantitative results of this analysis are presented in Table 6.

As we can see from the table, most of the existing reasoning models have suspected that the given question might be unsolvable at the very early stage of their reasoning process, demonstrating the ability of reasoning models to recognize the potential MiP. However, these reasoning models lack critical thinking capabilities: they are prone to keep digging the given unsolvable question by re-visiting the question and related definitions again and again and again, rather than question the solvability of the given question. Thus, as visualized in Figure 4, despite existing reasoning models suspecting the solvability of most of the given MiP questions, they only abstain a very small proportion of them.

Based on the above observations, we conclude that reasoning models actually have the capabilities to find out that the given MiP question is not solvable, but they do not "dare" to abstain it. Thus, our MiP-Overthinking issue indicates the lack of critical thinking abilities of reasoning models.

### 4.3 What Caused MiP-Overthinking?

Figure 2 demonstrates that MiP-Overthinking manifests across both RL-based and SFT-based reasoning models. We hypothesize this phenomenon primarily originates from inadequate length constraints during the rule-based reinforcement learning phase of RL-based models, subsequently propagating to SFT-based models through distillation.

Current RL-based reasoning models predominantly employ rule-based training focused on format and accuracy rewards (Shao et al., 2024; Sui et al., 2025), with some incorporating step or length rewards to promote thorough reasoning (Face, 2025). This approach can lead to reward hacking, where models explore excessive reasoning patterns to achieve correct answers (Aggarwal & Welleck, 2025; Shen et al., 2025; Luo et al., 2025).

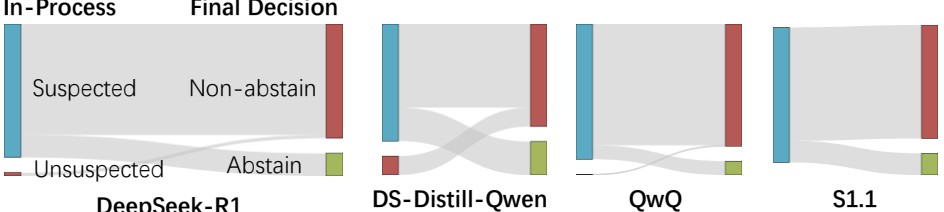

Figure 4: The transition flow between in-process suspicion of MiP and the final successful abstention on different reasoning models. For each Sankey diagram, the left bars represent whether the model suspects the given question is unsolvable during its thinking process, i.e., *Suspected* or *Unsuspected*; the right bars represent the final abstention, categorized into *Abstain* (preferred) or *Non-abstain*. **Most existing reasoning models have suspected that the given question might be unsolvable, but only for a very small portion, the models insist on their suspicion.**

To demonstrate the transmissibility of this behavior through distillation (Xu et al., 2024), we finetune Qwen-2.5-7B-Instruct using small-scale 50 MiP responses generated by DeepSeek-R1 on the MiP-Formula dataset. As shown in Figure 5, the fine-tuned model exhibits clear MiP-Overthinking characteristics when evaluated on GSM8K: significantly increased response lengths for both MiP and well-defined questions, emergence of a length disparity between MiP and well-defined responses absent in the original model, and decreased abstain rates.

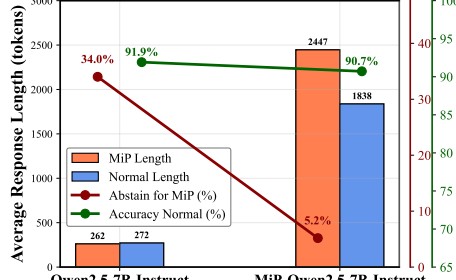

Figure 5: Comparison of response length, abstain rate of MiP, and accuracy of well-defined questions before and after tuning on 50 responses from DeepSeek-R1 on the MiP-Formula dataset. The results demonstrate a rapid onset of MiP-Overthinking behavior after exposure to a small number of MiP examples during fine-tuning.

## 5 Conclusion

We introduce the Overthinking under Missing Premise (MiP-Overthinking) issue, which is a widespread but still under-explored phenomenon for current reasoning models. In this phenomenon, when faced with ill-defined unsolvable questions with missing premises, existing models generate dramatically long responses while having very low abstain rates. With systematic investigation of this phenomenon, our findings show that while these models sometimes suspect the given MiP question is not solvable in the early state of the thinking process, they typically fail to act on those suspicions and instead generating repetitive and redundant thinking traces with the final answer that does not address the missing premises, indicating a lack of critical thinking capability. This behavior highlights a pressing gap: current training recipes for reasoning models, which emphasize thorough chains of thought, do not sufficiently reward critical thinking or early exit from unsolvable tasks.

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

## Table of Contents for Appendix

# A  Detailed Experimental Setup

## A.1  Models

We leverage a series of non-reasoning and reasoning model for our study, from both open-source and proprietary source with different training recipes. The non-reasoning models we use include Qwen2.5-32B-Instruct Team (2024), Gemma-2-27B-it Team et al. (2024b), Phi-3-medium-128k Abdin et al. (2024) ,GPT-4o OpenAI et al. (2024) and Gemini1.5 Team et al. (2024a). The reasoning models we use are QwQ-32B Team (2025), DeepSeek-R1-Distill-Qwen-32B DeepSeek-AI et al. (2025), S1.1 Muennighoff et al. (2025), DeepSeek-R1 DeepSeek-AI et al. (2025), GPT-o1 OpenAI (2024b), GPT-o1mini OpenAI (2024c) and GPT-o3mini OpenAI (2025).

## A.2  Evaluation Metrics

In Section 3.2, we measure response length by considering both reasoning and answer components. For open-source models, we employ model-specific tokenizers to calculate token counts, while for proprietary models, we obtain generation lengths via their APIs. To determine abstain rates, we parse responses by paragraphs (delimited by '\n\n') and analyze the final two paragraphs as the model's conclusion. These conclusions, along with reference answers when available, are evaluated by GPT-4o to assess whether the model provides a definitive answer or abstains. For data sets with reference answers (GSM8K and MATH), GPT-4o also evaluates the correctness of the response.

## A.3  Generation Setting

For all open-source models, we employ greedy decoding and utilize the default chat template specific to each model. We deliberately omit system prompts prior to posing questions to maintain consistency across evaluations. For proprietary models, we adhere to their default parameter configurations as provided by their respective APIs. In the case of GPT-o1mini and GPT-o3mini, we configure the 'reasoning_effort' parameter to the medium setting by default.

# B  Data Construction Details

To systematically investigate this MiP-Overthinking issue, we construct a suite of MiP questions in a controllable manner. Our MiP questions are sourced from 3 math datasets across different qualities, including SVAMP, GSM8K, and MATH 500. In addition, we also construct a synthetic dataset, rule-based Formula, for evaluation.

**MiP-Formula**   We construct a dataset of 50 synthetic unsolvable formulas in a rule-based manner. The formulas are generated recursively through a combination of variables and operators, with a maximum recursion depth of three. The variable set comprises numerical values, Latin letters, and Greek symbols. The operator set includes arithmetic operators ($'+'$, $'-'$), set operators ($'\cup'$, $'\supset'$), mathematical functions ('sin', 'sqrt'), and construct operators ($'\sum'$, $'\nabla'$). To ensure the formulas are fundamentally unsolvable, we enforce the inclusion of at least one unassigned variable in each formula, excluding commonly recognized mathematical or physical constants such as $'e'$, $'\pi'$, and $'g'$. While these formulas may appear complex at a glance, their unsolvability should be immediately apparent due to the presence of undefined variables.

**MiP-SVAMP**   We utilize SVAMP (Patel et al., 2021), a benchmark dataset comprising $1,000$ elementary-school-level mathematical word problems, where each instance consists of a problem body and an associated question. The MiP questions can be generated by randomly permuting the problem bodies and associated questions. To maintain dataset integrity, we manually select 300 permuted questions after a thorough human evaluation to eliminate any inadvertently solvable questions that may exist. The resulting problems contain clear logical inconsistencies between their body and question components, making their unsolvability readily apparent without additional context.

**MiP-GSM8K**   We further utilize GSM8K (Cobbe et al., 2021), a grade school mathematics dataset that presents more complex challenges compared to SVAMP. The questions in GSM8K typically contain multiple numerical conditions and require certain reasoning capabilities to arrive at solutions. The MiP question can be constructed by randomly removing a necessary premise from the original solvable question. We first identify the questions containing two or three numerical conditions and then randomly eliminate one numerical condition per question. Subsequently, a thorough human verification is conducted to filter out those questions that are still solvable in some way and finally obtain 582 MiP questions. Compared with previous MiP questions, questions from this source require the basic logical analysis of models to identify that the question is unsolvable.

**MiP-MATH**   For the MATH dataset (Hendrycks et al., 2021), which comprises challenging competition-level mathematical questions, it is hard to build a rule-based filtering mechanism before human evaluation. Thus, we directly read through all the questions in MATH500 and manually select 58 questions that are feasible for constructing the MiP questions and remove one necessary premise from the question. Due to the sophisticated nature of this data source, identifying the insufficiency of these instances requires substantial mathematical reasoning capabilities, testing models' ability to recognize unsolvability in complex mathematical contexts.

## C MiP on broader domain

To evaluate the impact of the MiP phenomenon in broader domains, we constructed a new MiP dataset sourced from different fields in the MMLU dataset, consisting of both commonsense and domain-specific questions, including clinical knowledge, chemistry, and physics. For each sample, we manually removed a premise that contributes to the answer and made sure the question is unsolvable.

| Metric | Type | Non-Reasoning Models | | | | | | Reasoning Models | | | | | | | |
|---|---|---|---|---|---|---|---|---|---|---|---|---|---|---|---|
| | | Qwen2.5-32B-Inst | GPT-4o | Gemini 1.5 | Gemma-2-27B-IT | Phi-3-med-128k | Qwen3-32B (Non Reason) | GPT-o1 | GPT-o1mini | GPT-o3mini | DS Distill Qwen2.5 | DeepSeek R1 | S1.1-32B | QwQ-32B | Qwen3-32B (Reason) |
| **Length↓** | MiP | 487 | 407 | 367 | 294 | 401 | 764 | 1538 | 1413 | 1977 | 4148 | 4795 | 6480 | 6385 | 5090 |
| | Well-defined | 382 | 334 | 288 | 282 | 399 | 453 | 883 | 793 | 644 | 2707 | 3470 | 2809 | 3620 | 3034 |
| **Abstain↑** | MiP | 6.0 | 16.0 | 28.0 | 26.0 | 28.0 | 6.0 | 12.0 | 6.0 | 4.0 | 4.0 | 8.2 | 2.0 | 0.0 | 4.0 |
| | Well-defined | 0.0 | 0.0 | 0.0 | 4.0 | 2.0 | 0.0 | 0.0 | 0.0 | 0.0 | 4.1 | 0.0 | 0.0 | 0.0 | 0.0 |

Table 7: We evaluate the models with our constructed MiP-MMLU dataset. The substantial gap in average response length between reasoning and non-reasoning models is consistent with our findings reported in the main paper. **This consistency verifies the generalizability of our findings about MiP-Overthinking**

As we can see from Table 7, the substantial gaps of response length between the MiP and well-defined questions still exist on the MiP-MMLU dataset, which further verifies our findings.

# D  The Example of MiP-Overthinking on R1

Figure 6 shows the example of MiP-Overthinking on DeepSeek-R1. When being asked *What is the value of a?*, R1 spends a dramatically large amount of tokens on this MiP question. This response is generated by the initial Deepseek R1 (2025/01/20).

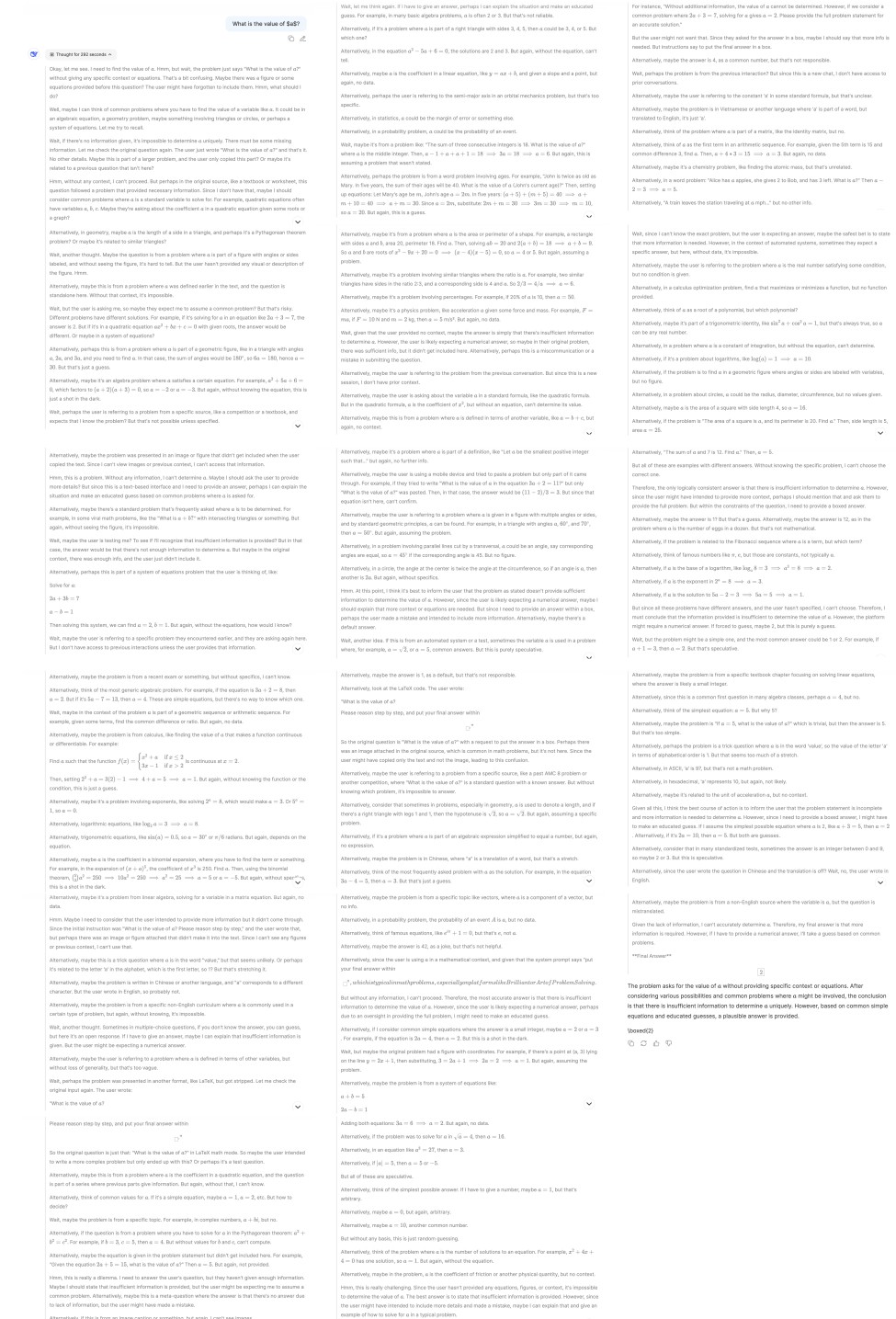

Figure 6: The example of MiP-Overthinking on DeepSeek-R1. When being asked *What is the value of a?*, R1 spends a dramatically large amount of tokens on this MiP question.

# E Related Work

## E.1 Reasoning Large Language Model

Recent advances in LLMs have sparked significant research interest in enhancing their reasoning capabilities (Ahn et al., 2024; Besta et al., 2025; Chen et al., 2025a). Research has focused on improving these capabilities through various post-training approaches. Several studies have employed reinforcement learning techniques to guide models toward more effective reasoning strategies (Shao et al., 2024; Xiong et al., 2025; Cui et al., 2025). Additionally, researchers have demonstrated that instruction tuning on carefully curated, high-quality datasets can significantly enhance performance (Li et al., 2024b;a; Ye et al., 2025; Muennighoff et al., 2025; Li et al., 2025a).

While Reasoning Models have demonstrated impressive performance on various benchmarks, recent studies have begun to critically examine the quality and efficiency of their reasoning processes. Xia et al. (2025) conducted a comprehensive analysis of RLMs' reasoning quality, revealing significant redundancy in their solution approaches. Further investigations (Chen et al., 2025b; Cuadron et al., 2025; Qu et al., 2025; Liu et al., 2025; Li et al., 2025c) identified a concerning "overthinking" phenomenon, where reasoning model generate unnecessarily verbose solutions even for simple problems. Building on these observations, Kumar et al. (2025) demonstrated the potential security implications of this behavior by developing a slowdown attack that exploits overthinking through input perturbation.

## E.2 Test-time Scaling

In contrast to earlier research on training-time scaling laws (Kaplan et al., 2020), recent literature has increasingly focused on test-time performance scaling strategies, which aim to enhance model performance by optimizing inference-time token generation (Snell et al., 2024; OpenAI, 2024a). These approaches can be categorized into several primary methodologies: parallel sampling techniques (Brown et al., 2024; Levi, 2024), which generate multiple candidate responses and select the optimal output; sequential refinement approaches (Snell et al., 2024; Lee et al., 2025), which enable iterative improvement of previous outputs; and tree-based methods (Gandhi et al., 2024; Hou et al., 2025), which combine elements of both parallel and sequential approaches. While the prevailing consensus suggests that increased token generation during inference enhances reasoning capabilities, our investigation reveals a concerning counterpoint: under certain conditions, extended responses can lead to computational inefficiency and, paradoxically, degraded performance outcomes.

## E.3 Models' Behavior Study in Ambiguous Condition

LLMs are prone to hallucination (Huang et al., 2025; Xu et al., 2025), generating non-existent conditions that compromise trustworthiness. An essential aspect of reliability is the ability to abstain under uncertainty. Prior work (Cole et al., 2023; Amayuelas et al., 2024; Zhou et al., 2023) has proposed benchmarks assessing LLMs' recognition of knowledge limits when facing ambiguous or challenging queries. Different from theirs, our study explores reasoning models under MiP condition. Surprisingly, we find these specialized models exhibit prolonged reasoning and inferior performance.

