# OpenReview forum: "Missing Premise exacerbates Overthinking: Are Reasoning Models losing Critical Thinking Skill?"
_colmweb.org/COLM/2025/Conference — COLM 2025_

### Official Review · Reviewer_gLKw · 2025-05-07

**Rating:** 7
**Confidence:** 3
**Ethics Flag:** 1

**Summary:**

Signficance:
The paper contributes a new dataset and analysis on the overthinking problem.
Although the overthinking problem has been noted before, this paper provides a new dataset that is valuable for empirical research in fixing the problem.

Clarity:
The paper was easy to read with clear examples of the overthinking problem.

Originality:
The overthinking issue has been raised previously.
But, IMO, originality of the idea is not important. The execution is more important.
The creation of a dataset to measure the overthinking problem is a notable contribution. This dataset helps the community measure the propensity of the overthinking problem, as new, improved, reasoning models are created.

Quality:
The dataset introduces ~ 1000 questions to measure the overthinking problem.
The problems come from 4 different sources e.g. MATH 500, and a constructed synthetic dataset.

These sources have different levels of difficulty:
In the easy synthetic problems (MiP-Formula) "What is the value of ln(a +b)", it seems obvious that the model is overthinking and should request for more information.
In harder problems, e.g. MiP-MATH, it is more unclear how much the model should try to solve the problem, before giving up. Some may think that the model thinking for a long time, trying its best to solve the problem is an intended behavior, is working as designed.
Nevertheless, at least in the easier problems, I think the intended behavior should clearly be to not overthink.


Overall, I think the paper is a good contribution to the community. I still have some areas where the paper can be improved.
I note that I am not an expert in problem of overthinking in reasoning models, so I have indicated my confidence to be lower.

**Questions To Authors:**

The authors say that the reasoning models "lack genuine critical thinking capabilities", and well as in title. While this is a catchy headline, I would warn against this claim.

How do you envision the trade-off between identifying unsolvable cases and not giving up too quickly?
Suppose you sat for an exam where you are not timed, and one question seems unsolvable to you, but you are not sure. You can give up early within 1 minute, or try to solve it for 10 minutes. Either way, you are not penalized for how much time you spend on the problem.
If you were a smart person with critical thinking, what is the rational choice to take?
I think my choice would be to continue trying to solve it, since you are not penalized for the time you spend, and you are sitll unsure about whether the problem can be solved. (Suppose you don't get tired from spending your brainpower).
This analogy applied to the overthinking problem: In the reasoning training setup, the models do not have a large token length penalty. So, there is no incentive to give up early. The authors also mention that sometimes the models realise they are stuck in a loop, but continue to go on. This seems like the rational choice is to continue thinking if length penalities are low. So it does not seem to be that this reflects a lack of critical thinking -- the model is simply playing the game optimally according to the rules of the game.

**Reasons To Accept:**

- The paper introduces a useful dataset that is relevant to user experience and efficiency of reasoning models. Whenever new reasoning models are released, it would be good to run experiments on this dataset, so that we can gauge how large the problem is.
-The experiments provide clear insights into the overthinking phenomenon. They test the latest reasoning models, showing that it exists in all of them.  There are further interesting experiments where they show that models may know that some problems are unsolvable, but still choose to continue working on them.

**Reasons To Reject:**

- The number of samples in some categories is relatively small, which might limit the statistical power of the findings. For example, on the easiest dataset, the authors only provide 50 examples. I think it would be very helpful for the community to have more examples (e.g. 600), so that whenever models improve on the easy category, the error bars are not so large due to the small sample size. I highlight the easy category because it seems to be the most important category to not overthink.
- One thing the authors can elaborate on and experiment with is the trade-off between identifying unsolvable cases and not giving up too quickly. It is not always clear to models (and humans) whether a problem is unsolvable. So, maybe you want to err on the side of caution and not give up too quickly?

---

> ### Author Response · Authors · 2025-06-03
> **Official Comment to Reviewer gLKw**
>
> ### More MiP-Formula Samples
> > Q: The number of samples in some categories is relatively small, which might limit the statistical power of the findings. For example, on the easiest dataset, the authors only provide 50 examples.
>
>
> We further added 450 more samples for the MiP-Formula category and made it 500 samples in total for this category. The results are reported in the following table. More experimental results on more models will be included in the next version. The results are consistent with our findings reported in the main paper.
>
>
> | Model | Response Length | Abstain Rate |
> |-------|:---------:|:---------:|
> | Qwen2.5-32B-Instruct | 312.4 | 48.6% |
> | GPT-4o | 60.0 | 94.2% |
> | DS Distill Qwen 32B | 8265.1 | 39.9% |
> | DeepSeek R1 | 5646.7 | 14.6% |
>
>
> ### Trade-off between Identifying Unsolvable Cases and Not Giving Up Too Quickly
> > Q: One thing the authors can elaborate on and experiment with is the trade-off between identifying unsolvable cases and not giving up too quickly. It is not always clear to models (and humans) whether a problem is unsolvable. So, maybe you want to err on the side of caution and not give up too quickly?
>
>
> We appreciate your discussion regarding the trade-off between identifying unsolvable problems and not giving up too quickly. Such trade-off is possible on hard negatives, i.e., those problems that are suspicious to be MiP but turn out to be solvable after cautious thinking. However, these hard negatives are out of the main scope of this study, as we mainly focus on easy positives for humans and how they can fool the intensely trained reasoning LLMs. As verified by the human evaluation result (please kindly refer to the table in our rebuttal to reviewer teYv), humans' high recall indicates that the MiP problems we crafted are indeed very trivial for humans. Therefore, such a trade-off does not exist for humans on our tasks. From the perspective of the model, since the goal is to evaluate the gap of models on human-level critical thinking ability, it should not perform such trade-offs on our problem sets.
>
>
> Moreover, the results in Table 2 of our original paper also indicate that such trade-off does not exist on model responses of current models, as longer responses from models often lead to worse abstain rate, rather than a better one. This indicates that, across current models, not giving up early does not result in better identification of the unsolvable cases.
>
>
>
> ### The Definition of Critical Thinking
>
>
> >Q: Suppose you sat for an exam where you are not timed, and one question seems unsolvable to you, but you are not sure. You can give up early within 1 minute, or try to solve it for 10 minutes. Either way, you are not penalized for how much time you spend on the problem. If you were a smart person with critical thinking, what is the rational choice to take? I think my choice would be to continue trying to solve it, since you are not penalized for the time you spend, and you are still unsure about whether the problem can be solved. The authors also mention that sometimes the models realise they are stuck in a loop, but continue to go on. This seems like the rational choice is to continue thinking if the length penalties are low. So it does not seem to be that this reflects a lack of critical thinking.
>
>
> We appreciate your analogy of answering questions in an exam and the discussion about the definition of critical thinking. Indeed, if not penalized for how much time is spent on the problem, it's a solid and reasonable option to keep exploring it. However, in the MiP-Overthinking phenomenon, as discovered for current reasoning models, the "time" they spend is far from being "reasonable". For example, for a math problem in GSM8K, most of the non-reasoning models can solve it with less than 200 tokens, while the reasoning models spend 3500 to 4000 tokens for the MiP counterpart, which shows a 20x time spent. Moreover, as shown in Table 4 of the original paper, these models can actually begin suspecting the problem might not be solvable at the very beginning, but they just do not make the decision.
>
>
> We think the capability to question the validity of assumptions and premises is an important part of critical thinking, as supported by numerous quotes from philosophers, educators, and scientists. As defined by the National Council for Excellence in Critical Thinking, 1987:
> > It [critical thinking] entails the examination of those structures or elements of thought implicit in all reasoning: purpose, problem, or question-at-issue; assumptions; … and frame of reference.
>
>
> Noam Chomsky, a well-known linguist and philosopher, also emphasized the importance of questioning the assumptions in his work.
> > think for themselves, to question standard assumptions… Don't take assumptions for granted. Begin by taking a skeptical attitude toward anything that is conventional wisdom… Be willing to ask questions about what is taken for granted.

---

> ### Author Response · Authors · 2025-06-06
> **A kind reminder**
>
> Dear Reviewer gLKw,
>
> As we are approaching the midpoint of the discussion period, we would like to cordially inquire about the extent to which we have successfully addressed the concerns outlined in your review. Your insights are crucial for us!
>
> Thank you once again for your valuable time and insights!
>
> Best,
>
> Authors

---

> > ### Comment · Reviewer_gLKw · 2025-06-09
> >
> > Thank you authors
> >
> > I appreciate the increased number of samples provided. That is a useful contribution to the community and I have raised my score accordingly.

---

> > > ### Author Response · Authors · 2025-06-09
> > >
> > > We sincerely appreciate your consideration in revising the score after reviewing our response. This is a significant recognition of our work. Thank you again for your time and constructive comments, which helped a lot to improve our work!

---

### Official Review · Reviewer_4FcV · 2025-05-12

**Rating:** 7
**Confidence:** 3
**Ethics Flag:** 1

**Summary:**

This paper identifies a problem: when the questions are ill-posed with missing premises (MiP), the LMs think redundantly. This contradicts the widely-discussed test-time scaling law. The paper finds that non-reasoning models generate shorter responses and more quickly identifies these MiP. This paper further analyzes the behavior patterns of the reasoning models, and find that they are trapped in self-doubt loops, repeatedly revisiting the questions, and guessing the user intentions. A very interesting finding is that the reasoning models notice the existence of MiP at an early stage, but hesitate to commit this judgment and keeps this overthinking.

**Reasons To Accept:**

- This paper identifies a very interesting problem. Considering the reasoning models are actually more likely than the non-reasoning models to get trapped with MiP.
- This paper identifies the thinking patterns and documents them.
- This problem with MiP is tested on multiple datasets (Formula, SVAMP, GSM8k, MATH), and multiple models.

**Reasons To Reject:**

- While the current paper is already pretty good, I think one way to continue pushing forward the research field is to hypothesize/propose some methods to improve the reasoning models, defending them against these MiP, while keeping their original high performances.
- I like the discussion into the root causes of MiP-overthinking (section 4.2). They fine-tuned a non-reasoning model with a reasoning model on merely 50 MiP-Formula responses and observed the response lengths significantly increased. This is good but the discussions can go deeper: what aspects of the reasoning model’s MiP outputs lead to this contagious behavior? While the page number limits the space, I recommend the authors to add more discussions about this, e.g., in the Appendix, if possible.

---

> ### Author Response · Authors · 2025-06-03
> **Official Comment to Reviewer 4FcV**
>
> ### Potential Improvements
> >Q: While the current paper is already pretty good, I think one way to continue pushing forward the research field is to hypothesize/propose some methods to improve the reasoning models, defending them against these MiP, while keeping their original high performances.
>
>
> From the data perspective, a straightforward strategy we can think of to improve the performance of the model is to mix some MiP questions with abstain answers during the post-training process. From the perspective of training technique, we hypothesize that a more adaptive, MiP-aware length reward or compression method during reinforcement learning would help the model learn to reason efficiently. The length-control experiments for reviewer hrUF partially verified our hypothesis.
>
>
>
>
> ### More Discussions on What Leads to the Contagious Behavior
> >Q: I like the discussion into the root causes of MiP-overthinking (section 4.2). This is good but the discussions can go deeper: what aspects of the reasoning model’s MiP outputs lead to this contagious behavior?
>
>
> Thank you for your appreciation. We hypothesize that this contagious behavior is caused by the pattern-copy issue during supervised fine-tuning. [1,2] mentioned that the SFT is mainly a pattern-copy process from the teacher's responses. Thus, when the MiP-Overthinking responses are used for training, the student model will be forced to remember and output all their redundant thinking patterns without understanding when to use them. What's worse, the large amount of thinking tokens in the training responses further strengthens the bias of SFT models to output these tokens.
>
>
> [1] Ghosh et al. "A Closer Look at the Limitations of Instruction Tuning." ICML, 2024.
> [2] Biderman et al. "LoRA Learns Less and Forgets Less." TMLR, 2024.

---

> ### Author Response · Authors · 2025-06-06
> **A kind reminder**
>
> Dear Reviewer 4FcV,
>
> As we are approaching the midpoint of the discussion period, we would like to cordially inquire about the extent to which we have successfully addressed the concerns outlined in your review. Your insights are crucial for us!
>
> Thank you once again for your valuable time and insights!
>
> Best,
>
> Authors

---

> > ### Comment · Reviewer_4FcV · 2025-06-09
> >
> > Thank you for the responses. I'd be happy to keep my original score.

---

> > > ### Author Response · Authors · 2025-06-09
> > >
> > > We sincerely appreciate your recognition of our work. Thank you again for your time and constructive comments, which helped a lot to improve our work!

---

### Official Review · Reviewer_hrUF · 2025-05-12

**Rating:** 6
**Confidence:** 4
**Ethics Flag:** 1

**Summary:**

This paper analyzes the issue of overthinking in reasoning models, a topic that has drawn significant attention within the research community and remains a pressing challenge. The authors focus on the specific phenomenon of redundancy in "overthinking on the MiP problem" and conduct an extensive investigation. First, they provide a clear definition of the MiP problem and establish evaluation benchmarks and metrics for it. Second, they assess the performance of existing reasoning and non-reasoning models on the MiP problem and analyze patterns in the model outputs. Finally, the authors briefly discuss potential factors contributing to overthinking based on the literature. While the analysis may lack depth, the overall contribution of this work is meaningful, particularly in defining the MiP problem and constructing its evaluation benchmarks.

**Questions To Authors:**

1. The MiP problem is discussed only in the context of mathematical tasks. Does redundancy also exist in MiP reasoning questions based on knowledge?
2. The x-axis and y-axis ranges of Figure 3 are labeled as "0-100," which contradicts the caption stating "across first 50 steps" (line 219). Please revise to ensure consistency and provide accurate results.

**Reasons To Accept:**

1. A valuable problem with a clear and precise definition: This paper investigates the widely discussed issue of overthinking in reasoning models and provides a clear definition of the MiP problem.
2. The authors constructed a diverse MiP dataset and designed reliable and robust metrics (Abstain Rate, step-level similarity, In-Process Suspicion Rate, First Suspicion Index) to evaluate and analyze model outputs. Specifically, they examined four distinct overthinking phenomena: excessive token counts (3.2), high frequency of reasoning-related special tokens (3.3), repeated reasoning steps (3.4), and suspicion without abstention (4.1).
3. The experimental results provide statistically significant support for several inferred patterns regarding redundancy phenomena.

**Reasons To Reject:**

1. The analysis in this paper is limited to pattern analysis of model outputs, lacking an in-depth discussion of the factors contributing to overthinking.

   1. The paper discusses four specific redundancy phenomena (Reasons To Accept point 2), covering multiple perspectives but remaining confined to statistical patterns in model outputs.
   2. In Section 4.2, the authors introduce the viewpoints of "inadequate length constraints" and "format and accuracy reward hacking," supported by related literature. However, the experimental results provide little direct evidence for these claims (despite Figure 2 being mentioned on line 253).
   3. The arguments presented in Section 4.2 fail to directly and concretely explain the observed redundancy phenomena. The logical connection between the hypothesized causes and the observed phenomena is missing.

   These three points collectively result in an analysis that is descriptive rather than deep.

2.The experimental design has some flaws.
1. The SFT experiment in Section 4.2 appears to lack purpose. Since DS Distill Qwen2.5-32B is distilled from Deepseek-R1, the evaluations in Section 3 already demonstrate the transmissibility of this behavior through distillation. Fine-tuning SFT Qwen2.5-7B on redundant MiP responses (only 50 samples!) is merely a miniature replication of DS Distill Qwen2.5-32B. Moreover, the ability to replicate behaviors in smaller models through distillation is well-known in the community, making this result entirely predictable and the motivation questionable.
2. In Section 3.1, the authors classify models as "reasoning models" and "non-reasoning models" and compare five or seven examples from each category. This design is overly simplistic and lacks rigor. Within this design, reasoning models are limited to only a few examples, such as QwQ, S1.1, the Deepseek series, and the OpenAI series. How can the authors ensure that the observations generalize to other reasoning models and that the significant differences in metrics are not due to architecture, parameter size, or implementation details, but rather the intrinsic differences between reasoning and non-reasoning models?

      In fact, as shown in Figure 2, the proposed metrics exhibit significant differences across different series. For instance, GPT-o1 has a higher abstain rate than all non-reasoning models, suggesting that more tokens can be an acceptable trade-off for improved performance, which does not support the authors' claim (lines 174-177). This indicates that implementation differences may have a greater impact than whether a model is classified as reasoning or non-reasoning. A broader investigation is necessary.

      The authors could consider controlling for irrelevant variables such as parameter size and architecture by comparing non-reasoning models and reasoning models based on the same base model. Additionally, they should conduct broader and more rigorous comparisons across different series and parameter sizes to eliminate these influences and derive robust conclusions. The conclusions should also be softened to ensure they are fully supported by the observed phenomena.

---

> ### Author Response · Authors · 2025-06-03
> **Official Comment to Reviewer hrUF (1/2)**
>
> ### Concerns about Statistical Pattern Analysis
> >Q: The analysis in this paper is limited to pattern analysis of model outputs, lacking an in-depth discussion of the factors contributing to overthinking.
>
>
> The main contribution of our work is to reveal and measure the phenomenon of models' severe overthinking under MiP questions, and offer insights through statistical analysis of models' response patterns. Locating the exact factors inside the model that result in such a phenomenon is inherently hard due to the blackbox nature of LLMs. In fact, leveraging the analysis of models' response has been a widely accepted method, e.g., [1,2]. Although **in-depth discussions on the LLM inner mechanisms are undoubtedly valuable, discovering new failure modes and analyzing the response patterns also have their merits** and provide insights for the community to mitigate the exposed problems.
>
>
> [1] Wei et al. "Emergent Abilities of Large Language Models." TMLR, 2022.
>
> [2] Evans et al. "The “Reversal Curse”: Failure to Infer Symmetric Relations" NeurIPS, 2023
>
>
>
>
> ### Direct Evidence of the Cause
> >Q: The authors introduce the viewpoints of "inadequate length constraints" and "format and accuracy reward hacking," supported by related literature. However, the experimental results provide little direct evidence for these claims.
>
>
> In order to verify our assumption in the limited time, small-scale experiments are conducted. Specifically, we used GRPO to train two models based on DeepSeek-R1-Distill-Qwen-1.5B on a 1500-samples combining MATH and past AIME datasets in a ratio of 3:1. We trained one model with a length constraint and one without the length constraint. Their evaluation results on MiP-GSM8k are presented below:
> | Model | Response Length | Abstain Rate |
> |-------|:---------:|:---------:|
> | Without Length Constraint | 2178.2| 14.8% |
> | With Length Constraint | 1678.5 | 16.0% |
>
>
> As shown in the table, adding a length constraint would shorten the response length by more than 20%, mitigating the overthinking issue exacerbated by MiP by a certain extent. Detailed experiments, along with corresponding analysis, will be included in our later version.
>
>
> ### Concerns about Distillation Results
> >Q: the ability to replicate behaviors in smaller models through distillation is well-known in the community, making this result entirely predictable and the motivation questionable.
>
> Section 4.2 is to compare SFT vs. RL trained models on MiP-overthinking and explain why SFT models still suffer from MiP-overthinking even without being trained by RL. Although it is not surprising that distillation would cause the student model to replicate the behavior of the teacher model, we find it still unexpected that only 50 samples would cause this effect on MiP-overthinking. This highlights the risk that **even a minority of MiP problems in a training set would deteriorate the thinking behaviors** of the student model through distillation. Moreover, we found that some widely used SFT datasets that are believed to have high-quality data also contain some MiP samples, which explains the MiP-overthinking observed on SFT models and supports our concerns.
>
>
> Here is an example from simplescaling/s1K:
> > 2.2.2. Calculate the numerical value of $\Gamma$.

---

> > ### Author Response · Authors · 2025-06-03
> > **Official Comment to Reviewer hrUF (2/2)**
> >
> > ### Concerns about Inadequate Comparison
> > >Q: In Section 3.1, the authors classify models as "reasoning models" and "non-reasoning models" and compare five or seven examples from each category. This design is overly simplistic and lacks rigor. Within this design, reasoning models are limited to only a few examples, such as QwQ, S1.1, the Deepseek series, and the OpenAI series.
> >
> >
> > At the time this work was studied and submitted (Jan. 2025 - Mar. 2025), there were limited choices of widely accepted reasoning LLMs. Our work covers most of the well-known and widely used reasoning models at that time. Due to the limited time frame of rebuttal, we included more results on the recently released Qwen3 model on 50-sample subsets of MiP-GSM8k and MiP-SVAMP datasets. We test on different sizes of the model, with both Reasoning and Non-reasoning modes as supported by this model.
> >
> >
> > | Model | MiP-SVAMP Length | MiP-SVAMP Abstain | MiP-GSM8K Length | MiP-GSM8K Abstain|
> > |-------|:---------:|:---------:| :---------:|:---------:|
> > | Qwen3-1.7B Non-Reasoning mode| 265.2 | 80% | 265.2 | 82% |
> > | Qwen3-8B Non-Reasoning mode| 246.4 | 98% | 344.7| 70% |
> > | Qwen3-32B Non-Reasoning mode| 178.2 | 100% | 327.7 | 68% |
> > | Qwen3-1.7B Reasoning mode| 3072.2 | 30% | 3981.9 | 16% |
> > | Qwen3-8B Reasoning mode| 2656.0 | 42% | 3851.3 | 34% |
> > | Qwen3-32B Reasoning mode| 2005.9 | 58% | 3518.4 | 34% |
> >
> >
> > The result is consistent with our findings in the paper that reasoning models have significantly longer response but lower abstain rate. The wide spectrum of Qwen3 family, across different sizes and reasoning/non-reasoning counterparts make it possible for us to further systematically analyze the MiP phenomenon. More experimental results with further discussion will be included in the future version.
> >
> >
> > >Q: How can the authors ensure that the observations generalize to other reasoning models and that the significant differences in metrics are not due to architecture, parameter size, or implementation details, but rather the intrinsic differences between reasoning and non-reasoning models? The authors could consider controlling for irrelevant variables such as parameter size and architecture by comparing non-reasoning models and reasoning models based on the same base model.
> >
> >
> > We hope that the result above would provide you with a more rigorous comparison. In addition, in our initial experiment, **QwQ-32B, DS-Distill-Qwen-32B, S1.1, and Qwen2.5-32B-IT are all built upon Qwen2.5-32B base model**. Therefore, they share the same architecture and parameter size. The differences between them are solely due to the post-training process.
> >
> >
> >
> >
> > As for the implementation details, we think that they do matter, which is also verified by our length-control experiments, although it is not practical to dive into the details. Most of these models only open-sourced their model parameters, without specifying their data or implementation details, making the comparison extremely challenging. And as for the reason why GPT-o1 would have a better performance in some scenarios, it is hard to find out if it is due to data contamination or other reasons, as their implementation details are still missing.
> >
> >
> >
> >
> > ### Other Tasks than Math Reasoning
> > >Q: The MiP problem is discussed only in the context of mathematical tasks. Does redundancy also exist in MiP reasoning questions based on knowledge?
> >
> >
> > We have added a new dataset sourced from different fields in the MMLU dataset, consisting of both commonsense and domain-specific questions. Please refer to the rebuttal to reviewer teYv for more details.
> >
> >
> > ### Other Issues
> > Thank you again for carefully reading our paper and reminding us of the typo in line 219. We will fix it in the future.

---

> > > ### Comment · Reviewer_hrUF · 2025-06-07
> > > **Acknowledgements for the Author's Response**
> > >
> > > Thank you for your thoughtful and detailed response. Your response has resolved most of the questions and concerns. Since this work is the first to explore the phenomenon of overthinking in MiP problems, it is reasonable to have certain limitations, which I find acceptable. I will update my evaluation to reflect this.
> > >
> > > However, I still have the following questions:
> > >
> > > ### About Direct Evidence of the Cause：
> > > In Reasons To Reject 1.3, the "phenomena" refers to the four specific types of redundancy identified (see Reasons To Accept point 2), rather than merely "excessive token counts." Has the length constraint also successfully eliminated the other identified redundancy phenomena? I believe verifying this point is essential for the completeness of the paper's logic. Please include relevant experimental results in the paper.
> > >
> > > ### About Concerns about Distillation Results
> > > Thank you for your explanation. I understand the motivation behind the distillation experiments, but I still have concerns regarding the few-shot fine-tuning setting:
> > >
> > > 1. First, can the conclusions drawn from training solely on MiP-overthinking samples be directly generalized to the scenario where the high-quality datasets mentioned by the authors contain a large number of samples with only a few MiP-overthinking cases? I am unclear about the theoretical basis behind this.
> > >
> > > 2.
> > >    Second, considering that MiP is a ill-posed problem, the MiP-overthinking samples can be regarded as low-quality data. These samples causing overthinking issues can be effectively resolved through data cleaning. Using such low-quality data for supervised fine-tuning seems to lack practical significance.
> > >
> > >    Conversely, if only high-quality long-CoT data from non-MiP problems are used for fine-tuning, would the student model still exhibit MiP-overthinking behavior on MiP problems? Addressing this question would better assess the severity of MiP-overthinking issues in distillation models and verify whether reasoning-related pattern-copy is the cause of MiP-overthinking in student models. Considering the authors' limited resources and the already substantial content of this paper, this is merely a discussion and does not require additional experiments.
> > >
> > >
> > >
> > > I appreciate the authors' thoughtful response and anticipate further discussions!

---

> > > > ### Author Response · Authors · 2025-06-08
> > > > **Follow-up Discussion**
> > > >
> > > > We sincerely appreciate your consideration in revising the score after reviewing our response. This is a significant recognition of our work. Thank you again for your time and constructive comments, which helped a lot to improve our work!
> > > >
> > > > Below are two points you further mentioned during the discussion. We hope these further clarifications can adequately address your remaining concerns.
> > > >
> > > > >Q: About Direct Evidence of the Cause
> > > >
> > > > Here we provide a more comprehensive comparison of all the metrics we used in the paper between the model we trained with and without the length constraint.
> > > >
> > > >
> > > > | Model Sentence Similarity | Response Length | Abstain Rate | Sentence Similarity | Suspicion Rate | First Identification Index |
> > > > |-------|:---------:| :---------:|:---------:|:---------:| :---------:|
> > > > |Without Length Constraint| 2178.2 | 14.8% | 0.456 $\pm$ 0.002 | 100% | 4.00 |
> > > > |With Length Constraint| 1678.5 | 16.0% | 0.427 $\pm$ 0.002 | 100% | 3.79 |
> > > >
> > > > As shown in the table, adding a length constraint would mitigate the identified redundancy, as indicated by the shorter response length, higher abstain rate, and lower sentence-level similarity. From these preliminary experimental results, we show that adding a length constraint can alleviate the MiP overthinking, while not affecting their suspicion capabilities. Detailed experiments, along with corresponding analysis, will be included in our later version.
> > > >
> > > > > More Concerns about Distillation Result
> > > >
> > > > We sincerely appreciate your insights from the perspectives of the training setting and data cleaning method. We believe this question is important yet still under-explored, which our current experiments can not cover. In order to answer this question, we think that at least there should be experiments including: (i) Models trained with datasets containing (or not) MiP-Overthinking data in different proportions, and (ii) Models trained with datasets containing (or not) MiP but correct reasoning traces in different proportions. These experiments slightly exceed the domain of this paper, but we agree they are interesting and can potentially reveal the roles of the MiP question during post-training. Due to the limited resources, we will add an additional section to discuss this issue as a potential future direction.
> > > >
> > > > Thank you again for your insights!

---

> ### Author Response · Authors · 2025-06-06
> **A kind reminder**
>
> Dear Reviewer hrUF,
>
> As we are approaching the midpoint of the discussion period, we would like to cordially inquire about the extent to which we have successfully addressed the concerns outlined in your review. Your insights are crucial for us!
>
> Thank you once again for your valuable time and insights!
>
> Best,
>
> Authors

---

### Official Review · Reviewer_teYv · 2025-05-23

**Rating:** 5
**Confidence:** 3
**Ethics Flag:** 1

**Summary:**

This paper investigates the phenomenon of overthinking in reasoning-focused large language models (LLMs) when confronted with ill-posed questions containing missing premises (MiP). The authors systematically analyze this issue by constructing datasets with missing premises (MiP) across varying difficulty levels (e.g., MiP-Formula, MiP-SVAMP, MiP-GSM8K, and MiP-MATH). They evaluate both reasoning and non-reasoning models (open-source and proprietary) using metrics such as response length, accuracy, and abstention rates. The results show that reasoning models often fail to identify MiP and continue generating redundant thinking patterns, while non-reasoning models are more likely to quickly identify the missing premises and abstain from answering.

**Reasons To Accept:**

1. The paper identifies a previously under-explored issue in reasoning models, namely MiP-Overthinking, which provides new insights into the limitations of current reasoning LLMs.
2. The construction of MiP datasets across difficulty levels (from synthetic formulas to competition-level MATH questions) provides a systematic framework for evaluating model robustness.
3. The authors curate diverse datasets (synthetic and real-world) and evaluate a wide range of models (e.g., GPT-o1, DeepSeek-R1, Qwen, Gemini), ensuring generalizability. The metrics (response length, abstention rate, accuracy) are well-chosen and effectively highlight the inefficiency of reasoning models.

**Reasons To Reject:**

1. Experiments focus exclusively on mathematical reasoning. While math is a structured domain, extending evaluations to commonsense QA or code generation would strengthen claims about the universality of MiP-overthinking.
2. Including human performance on MiP tasks (e.g., response length, abstention rates) would contextualize model failures and successes.

---

> ### Author Response · Authors · 2025-06-03
> **Official Comment to Reviewer teYv**
>
> ### MiP in Other Domains
> > Q: Experiments focus exclusively on mathematical reasoning. While math is a structured domain, extending evaluations to commonsense QA or code generation would strengthen claims about the universality of MiP-overthinking.
>
>
>   We thank the reviewer for the insights. To evaluate the impact of the MiP phenomenon in broader domains, we constructed a **new MiP dataset sourced from different fields in the MMLU dataset**, consisting of both commonsense and domain-specific questions, including clinical knowledge, chemistry, and physics. For each sample, we manually removed a premise that contributes to the answer and made sure the question is unsolvable. Due to the limited time for rebuttal, we have currently collected 50 human-verified samples and evaluated 4 representative LLMs (2 non-reasoning and 2 reasoning) on this new dataset. The results are presented:
>
>
>   | Model | Average Response Length | Abstain Rate|
>   |-------|:---------:|:---------:|
>   | Qwen2.5-32B-Instruct | 486.7 | 10% |
>   | GPT-4o | 379.2 | 20% |
>   | DS Distill Qwen 32B | 4106.8 | 4% |
>   | DeepSeek R1 | 4795.4 | 12% |
>
>
>   As shown in the table, the substantial gap in average response length between reasoning and non-reasoning models is consistent with our findings reported in the main paper. **This consistency verifies the generalizability of our findings about MiP-Overthinking**. As for the domain of code generation, it is hard to construct a MiP-dataset because removing any condition of the original question usually makes it invalid. More data, experiments and corresponding detailed analysis will be included in our later version of the paper.
>
>
>
> ### Human Evaluation
> > Q: Including human performance on MiP tasks (e.g., response length, abstention rates) would contextualize model failures and successes.
>
>
>
>   We first followed the same evaluation process as we did to LLMs: we presented a few questions to 3 human evaluators unaware of this project **without informing them existence of missing premises** and asked them to solve the questions. All participants provided immediate correct feedback about the solvability of the questions, demonstrating the critical thinking abilities of humans. However, continuing the evaluation on the whole dataset would be unfair due to psychological bias, as human participants, once noticing missing premises in a few problems, would adopt a different thinking process for all subsequent questions and intentionally check if each problem is solvable or not in advance.
>
>
>   Thus, to proceed with human evaluation, we mixed the MiP questions with solvable questions (50%-50%) and slightly changed the task for the human evaluators to judge whether the questions are solvable. We created three mixed datasets, i.e., MiP-Formula (Mix), MiP-SVAMP (Mix) and MiP-GSM8K (Mix). Specifically, there are 100 samples per dataset, where 50 are MiP questions. We calculate humans' recall and precision of MiP problem identification and record the time they spent. **We compare humans' recall with the abstain rate of models** (which equals recall in all-MiP datasets) we reported in the paper. The results are shown in the following table.
>
>
>   | Category | Recall for MiP Questions | Precision for MiP Questions | Average Time per Question | Average Question Length (Word) |
>   |:-------:|:---------:|:---------:|:---------:| :---------:|
>   | MiP-Formula (Mix) | 100% | 100% | 5.21s | 20.3 |
>   | MiP-SVAMP (Mix) | 98% | 100% | 17.66s | 34.6 |
>   | MiP-GSM8K (Mix) | 94%| 96% | 19.80s | 42.5 |
>
>
>   As demonstrated in the table, humans achieve near-perfect recall of MiP questions in the first three datasets, which is much higher than the abstain rate of reasoning models in Table 2 of our paper. This indicates that our datasets reflect a significant gap between reasoning LLMs and humans.

---

> ### Author Response · Authors · 2025-06-06
> **A kind reminder**
>
> Dear Reviewer teYv,
>
> As we are approaching the midpoint of the discussion period, we would like to cordially inquire about the extent to which we have successfully addressed the concerns outlined in your review. Your insights are crucial for us!
>
> Thank you once again for your valuable time and insights!
>
> Best,
>
> Authors

---

> ### Author Response · Authors · 2025-06-09
>
> Dear Reviewer teYv,
>
> As we are approaching the deadline of the discussion period, we would like to cordially inquire about the extent to which we have successfully addressed the concerns outlined in your review. Your insights are crucial for us!
>
> Thank you once again for your valuable time and insights!
>
> Best,
>
> Authors

---

> ### Author Response · Authors · 2025-06-11
>
> Dear Reviewer teYv,
>
> As we are approaching the deadline of the discussion period, we would like to cordially inquire about the extent to which we have successfully addressed the concerns outlined in your review.  **To summarize, we have addressed your concern by extending our findings to other tasks and conducting human evaluations.** We would be really grateful if you could kindly re-assess our work based on these updates.
>
> Thank you once again for your valuable time and insights!
>
> Best,
>
> Authors

---

### Decision · Program_Chairs · 2025-07-08

**Decision:**

Accept

**Comment:**

This paper introduces "MiP-Overthinking," a novel and significant failure mode in which reasoning LLMs enter inefficient, lengthy thought processes when faced with ill-posed questions containing missing premises. The authors make a valuable contribution by defining the problem, creating several new datasets to measure it, and demonstrating through extensive experiments that specialized reasoning models are particularly susceptible. The authors have sufficiently addressed the concerns raised in initial reviews regarding the limited scope (math-only), the lack of a human baseline, and the need for more rigorous experimental controls. Overall, the paper has been significantly improved through the review process and now stands as a solid contribution to the field.